# Scalable Real-World Robot Data Generation via Compositional World Simulation

## Abstract

Recent advancements in foundational models, such as large language models and world models, have greatly enhanced the capabilities of robotics, enabling robots to autonomously perform complex tasks. However, acquiring large-scale, high-quality training data for robotics remains a challenge, as it often requires substantial manual effort and is limited in its coverage of diverse real-world environments. To address this, we propose a novel hybrid approach called **Compositional Simulation**, which combines classical simulation and neural simulation to generate accurate action-video pairs while maintaining real-world consistency. Our approach utilizes a closed-loop real-sim-real data augmentation pipeline, leveraging a small amount of real-world data to generate diverse, large-scale training datasets that cover a broader spectrum of real-world scenarios. We train a neural simulator to transform classical simulation videos into real-world representations, improving the accuracy of policy models trained in real-world environments. Through extensive experiments, we demonstrate that our method significantly reduces the sim2real domain gap, resulting in higher success rates in real-world policy model training. Our approach offers a scalable solution for generating robust training data and bridging the gap between simulated and real-world robotics.

## 1 Introduction

With the rapid advancements in foundational models, such as large language models OpenAI (2025); Touvron et al. (2023); Team et al. (2023) and world models OpenAI (2024); Bruce et al. (2024); NVIDIA (2025), there has been significant progress in the field of robotics Du et al. (2023); Yang et al. (2023). These innovations have enabled robots to autonomously perform increasingly complex tasks, opening the door to more capable and adaptable robotic systems. Data-driven paradigms have led to impressive results in various domains, yet robotics presents unique challenges compared to fields like language and video modeling. In particular, the need for manually collected video-action pairs poses a significant barrier. Unlike self-supervised learning techniques in other domains, acquiring large-scale data for robotics requires substantial human effort, which is both costly and insufficient for capturing the vast diversity of real-world environments.

While some researchers have addressed this issue by relying on large-scale human data collection, this approach remains expensive and limited in covering the full spectrum of real-world distributions. An alternative method is to leverage simulation to scale data collection Nasiriany et al. (2024); Mu et al. (2024); Qin et al. (2025), thus reducing the costs associated with real-world data acquisition. Classical simulators, such as MuJoCo Todorov et al. (2012) and Isaac Makoviychuk et al. (2021), offer the advantage of generating precise action-video pair data. These simulators use omniscient views, making it easy to generate vast amounts of data with diverse distributions. However, the performance gap between simulated and real-world environments often leads to poor joint training performance when directly transferring simulated data for real-world applications.

To bridge this gap, neural simulators Bruce et al. (2024); Yu et al. (2025); NVIDIA (2025) based on video generation models Blattmann et al. (2023); Wan et al. (2025); Zheng et al. (2024) have recently been proposed as a solution. These simulators generate corresponding video data from input trajectories or action signals, producing action-video pairs for training. Although the generated videos appear visually consistent with the real world, issues like hallucination—where videos lack physical consistency and lead to poor action control—undermine the quality of the generated data.

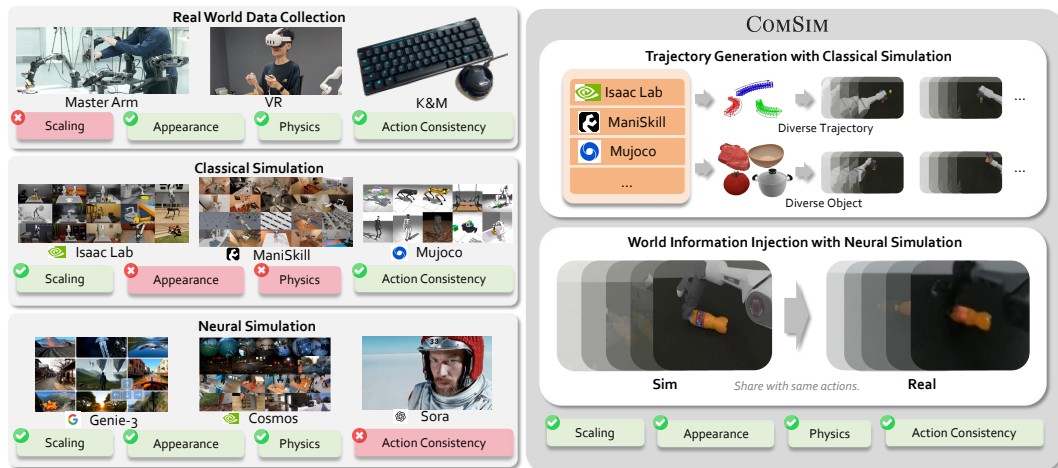

Figure 1: There are three main sources of real-world robotic data: (1) direct human collection, which yields high-quality samples but cannot scale; (2) classical simulators, which generate large datasets but suffer from appearance and physics gaps to reality; and (3) neural simulators trained on real data, which reduce these gaps but struggle with action-conditioned video generation, leading to weak action–video consistency. We introduce the concept of *Compositional Simulation*, a flexible and scalable approach that bridges the gap between classical simulation and real-world dynamics via compositional simulation.

In this work, we introduce the concept of **Compositional Simulation** shown in Fig 1, advocating for a hybrid approach that combines the strengths of classical simulation and neural simulation. This approach aims to generate accurate action-video pairs while ensuring that the videos are consistent with real-world dynamics. We propose a closed-loop real-sim-real data augmentation pipeline that utilizes a small amount of real-world data to create training datasets for policy models. These datasets are designed to cover a broader distribution of real-world scenarios.

The real-sim-real pipeline consists of two key steps. First, we collect a small set of real-world trajectory data and obtain the corresponding videos. In a classical simulation environment, we then replicate the same scenario, replaying the real-world trajectories to generate simulated videos. A video-to-video neural simulator is subsequently trained to transform the classical simulation videos into real-world videos, ensuring that the actions remain consistent. The second part of our approach involves generating a large and diverse set of action-video pairs through action primitives scheduling within the classical simulator. These pairs are then transformed into real-world representations using the trained video-to-video neural simulator, facilitating large-scale data augmentation for real-world applications. Our main contributions are as follows:

- *Concept & Paradigm.* We introduce the concept of *Compositional Simulation*, a flexible and scalable approach that bridges the gap between classical simulation and real-world dynamics via compositional simulation.

- *Data Pipeline & Model.* We propose a real–sim–real data augmentation pipeline that builds a neural simulator ensuring accurate and consistent action–video alignment while simultaneously mitigating the sim2real domain gap.

- *Experimental Results.* Extensive experiments demonstrate that *Compositional Simulation* substantially enhances the policy models by simultaneously increasing task success rates and achieving strong generalization across both spatial layouts and object variations.

## 2 RELATED WORK

### 2.1 ROBOTIC SIMULATION

Robotics simulation frameworks such as Isaac Lab Makoviychuk et al. (2021) and MuJoCo Todorov et al. (2012) are open-source, general-purpose simulators with GPU-parallel capabilities. Isaac

Lab and Brax Freeman et al. (2021) are closest to ManiSkill3: they ship ready-to-use environments for RL/IL and provide APIs for custom environment design. In contrast, frameworks like RoboCasa Nasiriany et al. (2024), Habitat Szot et al. (2021), AI2-THOR Kolve et al. (2017), Omni-Gibson Li et al. (2022), and RoboFactory Qin et al. (2025) emphasize predefined, logically structured APIs that standardize action interfaces, enable systematic domain randomization, and facilitate large-scale, programmatic dataset generation with correctness checks—allowing researchers to scale data and experiments reliably across tasks and scenes. ManiSkill3 uses the open-source SAPIEN Xiang et al. (2020) for GPU-parallel simulation. Despite these strengths—especially their clean action semantics and deterministic stepping that enforce logical action consistency—current simulators still fall short of real-world fidelity, with persistent gaps in appearance statistics, sensor characteristics, and contact/transfer physics (e.g., compliance and long-horizon object dynamics). This misalignment often yields policies that do not transfer robustly to real data and real dynamics.

## 2.2 ROBOT LEARNING IN MANIPULATION

Specialized policy architectures Chi et al. (2023); Ke et al. (2024); Liang et al. (2023; 2024; 2025); Wang et al. (2024); Wen et al. (2025); Ze et al. (2024) often excel on narrowly defined tasks yet struggle to carry over to new robot embodiments. In contrast, foundation models trained on million-scale, multi-robot corpora exhibit strong zero-shot transfer: RT-1 Brohan et al. (2022b) unifies vision, language, and action in a single transformer for real-time kitchen manipulation; RT-2 Brohan et al. (2023) jointly finetunes large vision–language models on web and robot data to support semantic planning and object reasoning; diffusion-based RDT-1B Liu et al. (2024) and $\pi$Black et al. (2024) learn diverse bimanual dynamics from over a million episodes. Vision–language–action systems such as OpenVLAKim et al. and CogACT Li et al. (2024), together with adaptations like Octo Octo Model Team et al. (2024), LAPA Ye et al., and OpenVLA-OFT Kim et al. (2025), further demonstrate efficient finetuning across robots and sensing modalities. Collectively, these results point to a data-driven bottleneck: robust cross-task and cross-embodiment generalization hinges on large, diverse, and high-fidelity datasets that faithfully capture real-world appearance, sensing, and physics.

## 2.3 WORLD SIMULATOR FOR ROBOTIC MANIPULATION

Scalable robot learning Bjorck et al. (2025); Brohan et al. (2022a); Zitkovich et al. (2023); Cheang et al. (2024); Lynch et al. (2023) depends on abundant, realistic data, yet collecting real-world trajectories via human demonstrations is slow and labor-intensive, limiting broad access. Generative video models Agarwal et al. (2025); Wu et al. (2023) offer a cost-effective way to synthesize policy training data. UniPi Du et al. (2023) and AVDC Ko et al. (2023) cast robot planning as text-to-video generation (AVDC further estimates inverse dynamics with a pretrained flow network); UniSim Yang et al. (2023) learns a unified real-world simulator across text and control inputs; RoboDreamer Zhou et al. (2024) targets compositional generalization via text parsing; and IRASim Zhu et al. (2024) performs trajectory-conditioned video generation but focuses on arm motion only. In this work, our world simulator turns action-consistent simulation trajectories into high-fidelity, real-style data.

# 3 COMPOSITIONAL WORLD SIMULATION

## 3.1 PROBLEM FORMULATION

In the context of robotic manipulation, collecting real-world data is often a challenging and resource-intensive task. Traditional methods leverage classical simulators Todorov et al. (2012); Makoviychuk et al. (2021); Gu et al. (2023) to train online reinforcement learning policies Schulman et al. (2017). These simulators generate large amounts of trajectory data by simulating various robot behaviors. Another approach Mu et al. (2024); Qin et al. (2025) utilizes pre-designed primitive functions, called via large language models (LLMs), to generate extensive trajectory data, thereby aiming to cover as much of the decision space as possible. These trajectories are commonly used for pre-training or joint training with real-world data.

Despite the large volume of video-action pairs generated, the disparity between the distributions of simulated and real-world data creates significant challenges. Let $\mathcal{D}_{\text{sim}} = \{(v_i, a_i)\}_{i=1}^{N}$ represent the dataset of video-action pairs collected from a classical simulator, where $v_i$ denotes the video frame

and $a_i$ the corresponding action. Similarly, let $\mathcal{D}_{\text{real}} = \{(v'_j, a'_j)\}_{j=1}^M$ represent the real-world dataset, where $v'_j$ and $a'_j$ are the video and action pairs from the real world. Directly training policies on the combined simulated and real data often fails to improve performance or generalization, as the domain gap between simulation and reality exacerbates this issue, leading to degraded policy performance in real-world settings. This gap is particularly evident in appearance and physics, where simulated data cannot fully capture the complexities of the real world.

An alternative method involves using video generation models as neural world simulators. These models generate data that is intended to be as close as possible to real-world distributions. However, video generation models suffer from inherent issues, such as hallucinations, 3D scene consistency, and inaccurate action control. As a result, the generated actions and corresponding videos do not align perfectly, making this data unsuitable for policy training.

To address these issues, we propose a compositional simulation approach. In this approach, we first collect a large number of trajectories in a classical simulator, $\mathcal{D}_{\text{sim}}$. These trajectories are then transformed into video representations using a pre-trained neural simulator $\mathcal{N}$, which maps the simulated data into the real-world distribution. Crucially, this process ensures that the generated data maintains action alignment with the original simulated trajectories. Formally, we aim to build a neural simulation function $\mathcal{N}(\cdot)$, such that:

$$\mathcal{N}(\mathcal{D}_{\text{sim}}) \approx \mathcal{D}_{\text{real}} \tag{1}$$

This neural simulation function $\mathcal{N}(\cdot)$ maps the simulated video-action pairs to a distribution that is as close as possible to the real-world data, ensuring that the generated action $a_i$ aligns with the original action $a'_j$, where $a_i \approx a'_j$. Additionally, the consistency of the 3D scene and the video quality must be maintained, addressing the inherent challenges in video generation models. Thus, we transform the simulated data $\mathcal{D}_{\text{sim}}$ to approximate the real-world distribution $\mathcal{D}_{\text{real}}$, while ensuring that the generated actions and videos are consistent with real-world expectations. By applying this compositional simulation approach, we can effectively utilize the large-scale data generated in simulators and adapt it to real-world environments, thereby mitigating the challenges posed by domain gaps in robotic manipulation tasks.

## 3.2 SIM2REAL NEURAL SIMULATION

To train the Sim2Real neural simulation that maps videos to real-world distributions while maintaining the correct actions, we need to construct a dataset composed of tuples $(\mathcal{V}_{\text{sim}}, \mathcal{V}_{\text{real}}, \mathcal{A})$, where $\mathcal{V}_{\text{sim}}$ and $\mathcal{V}_{\text{real}}$ represent the results of executing the same action in the classical simulator and the real world, respectively. In other words, $\mathcal{V}_{\text{sim}}$ and $\mathcal{V}_{\text{real}}$ share the same action sequence.

To build such a dataset, we need to create a simulation data collection platform that aligns strictly with the real-world data collection platform. As shown in Fig. 2, to establish this digital twin simulation environment, we performed alignment at three levels:

**Background and Object Alignment:** We first aligned the background and objects in the simulation, including their colors and sizes. The desktop and background colors in the classical simulator were aligned with those of the real-world data collection platform. Additionally, we applied a digital twin approach to assets to ensure visual consistency and set the size to match the real-world scale.

**Camera Calibration and Alignment:** We then calibrated and aligned the cameras to ensure that the camera parameters and poses in the real world were consistent with those in the classical simulation.

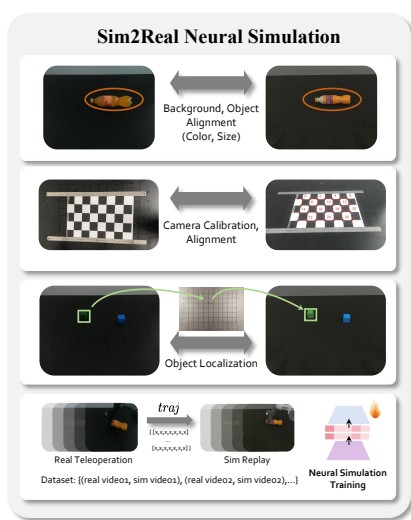

Figure 2: Alignment between real-world and simulation: trajectories collected in the real world are replayed in simulation to generate paired video data for training the sim-to-real neural simulator.

Figure 3: **Real World Deployment with *Compositional Simulation*.** Large volumes of $(\mathcal{V}_{\text{sim}}, \mathcal{A})$ pairs are collected from the classical simulator and transformed into corresponding $(\mathcal{V}_{\text{real}}, \mathcal{A})$ pairs, referred to as *Pseudo Real Data*. These data, together with a small amount of real-world data, are used to train policies with improved success rates and generalization.

**Object Position Alignment:** During task initialization, we localized the objects in the real-world scene and strictly transferred their position information into the classical simulator.

After performing the above alignments, we can collect data from the real-world simulation platform to obtain the pair $(\mathcal{V}_{\text{real}}, \mathcal{A})$. These action data are then replayed in the corresponding classical simulation environment to generate the tuple $(\mathcal{V}_{\text{sim}}, \mathcal{V}_{\text{real}}, \mathcal{A})$. We collected data for 10 tasks, resulting in 200 data pairs for training. To optimize the neural simulator for Sim2Real data generation, we aim to minimize the discrepancy between the simulated and real-world videos, while maintaining the correct action alignment. Since the actions in both $\mathcal{V}_{\text{sim}}$ and $\mathcal{V}_{\text{real}}$ are already aligned, we focus solely on optimizing the video consistency. The optimization objective is formulated as:

$$\mathcal{L}_{\text{sim2real}} = \mathcal{L}_{\text{video}}(f_{\mathcal{N}}(\mathcal{V}_{\text{sim}}, \theta), \mathcal{V}_{\text{real}}) \tag{2}$$

Where $\mathcal{L}_{\text{video}}$ measures the difference between the generated simulated video and the real-world video, and $\theta$ represents the neural simulator's parameters. By minimizing this loss, the neural simulator learns to generate videos that closely match the real-world distribution, while preserving the correct action alignment.

## 3.3 DATA GENERATION WITH RULE-BASED SIMULATION

To further scale up the data collection pipeline, we employ RoboTwin Chen et al. (2025), a SAPIEN-based Xiang et al. (2020) dual-arm manipulation simulation environment. It provides a rich library of digital assets and supports diverse trajectory distributions, making it well-suited for synthesizing large-scale visuomotor datasets. By systematically varying environmental conditions, object initialization states, and agent actions, we generate an extensive set of trajectories and corresponding videos that cover a broad spectrum of real-world task scenarios.

Specifically, we define a comprehensive set of interaction rules, referred to as action primitives, governing how agents and objects interact within the simulation. These primitives serve as the atomic building blocks of complex behaviors, capturing low-level manipulations (e.g., grasp, push, align) as well as higher-order skills (e.g., stack blocks). We curate a suite of RoboTwin tasks and adapt them to support richer interaction patterns and object configurations, enabling a broader spectrum of physical reasoning scenarios. To automate the generation of complex behaviors, we employ GPT-5 OpenAI (2025) to synthesize executable code composed of these action primitives, while integrating compositional constraints Qin et al. (2025) to ensure semantic correctness and physical feasibility. The action primitives encompass a variety of object types and interaction modalities, enabling diverse scenario generation. For each task, we construct a rich collection of trajectories $\tau_s$ spanning the action space, and carefully tune the primitive-based generation process to achieve comprehensive coverage. This allows us to traverse the global distribution of agent behaviors in the simulation, including different object initializations and heterogeneous object categories.

The resulting dataset comprises temporally synchronized camera observations, corresponding action and state sequences. These elements are strictly aligned at the behavioral level, ensuring that every visual frame is paired with its underlying control command. Although the trajectories and interactions in simulation are faithful to their intended semantics, the rendered appearance of the videos still differs from real-world imagery due to discrepancies in lighting, textures, and sensor noise. To bridge this domain gap, we pass the simulated observation streams $v_s$ through a neural simulator $\mathcal{N}$, which refines their visual characteristics while preserving the original dynamics and action consistency.

### 3.4 Real World Deploy with Compositional Simulation

As shown in Fig. 3, after training the neural simulator, we proceeded with the process outlined in Sec. 3.3 to collect a large number of $(\mathcal{V}_{sim}, \mathcal{A})$ pairs from the classical simulation. These data are then fed into the neural simulator, which transforms them into corresponding $(\mathcal{V}_{real}, \mathcal{A})$ pairs. We refer to these transformed data as *Pseudo Real Data*. Compared to the data produced by classical simulators, these Pseudo Real Data exhibit representations that are much closer to real-world data, with a reduced domain gap.

By using these Pseudo Real Data, which cover a broader distribution of scenarios, in conjunction with a small amount of real-world data collected from the actual environment, we can jointly train a robot policy. This approach significantly improves the performance and generalization capability of the policy. The specific experimental results are presented in the Sec. 4.2.

---

**Algorithm 1** Real World Deployment with Compositional Simulation

---

1: **Input:**
2:     Classical simulation data $(\mathcal{V}_{sim}, \mathcal{A})$, Real-world data $(\mathcal{V}_{real}, \mathcal{A})$
3: **Functions:**
4:     Neural Simulator $\mathcal{N}$, Video Transformation Function $f_{\mathcal{N}}$
5: **Hyperparameters:**
6:     Real-World Data Ratio $\alpha$
7: Initialize $D_{sim} \leftarrow \{\mathcal{V}_{sim}, \mathcal{A}\}, D_{real} \leftarrow \{\mathcal{V}_{real}, \mathcal{A}\}$       ▷ Initialize datasets
8: $P_{pseudo} \leftarrow \{\}$       ▷ Initialize Pseudo Real Data set
9: **for** each $(\mathcal{V}_{sim}, \mathcal{A}) \in D_{sim}$ **do**
10:     $P_{pseudo} \leftarrow P_{pseudo} \cup f_{\mathcal{N}}(\mathcal{V}_{sim}, \mathcal{A})$     ▷ Transform simulation data to Pseudo Real Data
11: **end for**
12: $D_{combined} \leftarrow \alpha \cdot D_{real} + (1 - \alpha) \cdot P_{pseudo}$     ▷ Combine Pseudo Real Data with Real Data
13: Train policy $\pi_{robot}$ using $D_{combined}$     ▷ Train robot policy using combined data
14: **Return:** Trained robot policy $\pi_{robot}$

---

## 4 Experiments

### 4.1 Sim2Real Transfer via Neural Simulation

**Baselines.** To validate the effectiveness of our proposed Neural Simulation in recovering real-world data distributions from simulation, we consider three variants: 1) Classical Simulation, denoting the raw simulation videos without neural refinement; 2) Zero-Shot, referring to the base model applied without any sim-to-real fine-tuning; and 3) Ours, the proposed Neural Simulation method capable of generating pseudo-realistic content. By contrasting these baselines, we perform an ablation study to empirically evaluate the ability of our method to bridge the discrepancy between simulation and reality. Specifically, we provide each model with a simulation video together with a sim-to-real instruction, and expect the model to generate a corresponding pseudo-real video. Our framework is built upon Stable Diffusion 1.5 Rombach et al. (2022) as the base model, augmented with a post-processing strategy Yang et al. (2024) to alleviate temporal discontinuities across frames.

**Quantitative Results.** For quantitative evaluation, we employ a set of widely used perceptual and structural similarity metrics (PSNR, SSIM, CLIP Score Hessel et al. (2021), LPIPS Zhang et al. (2018)), alongside distributional measures (FID Heusel et al. (2017), FVD Unterthiner et al. (2018)), to assess both the visual fidelity of the generated videos with respect to real-world videos and their temporal coherence throughout the frame sequence. Tab. 1 reports the quantitative results, where

Table 1: Comparison of the realism quality of generated videos across different methods.

|  | PSNR ↑ | SSIM ↑ | CLIP Score ↑ | LPIPS ↓ | FID ↓ | FVD ↓ |
|---|---|---|---|---|---|---|
| **Sim** | 18.973 | 0.7870 | 0.7699 | 0.3781 | 172.71 | 624.74 |
| **Zero-Shot** | 13.093 | 0.5487 | 0.7308 | 0.4756 | 219.74 | 1163.83 |
| **Ours** | **19.240** | **0.8114** | **0.8011** | **0.2644** | **145.70** | **488.82** |

"Sim" indicates the original simulation data from the classical simulator, "Zero-Shot" denotes the outputs generated by the untuned base model, and "Ours" corresponds to the pseudo-real videos synthesized by our proposed neural simulator. We observe that the base model, when used without any sim-to-real adaptation, is not only ineffective but also hinders the realism of generated videos. In contrast, our method achieves the best performance across all evaluation metrics, consistently yielding videos with high perceptual realism and thereby demonstrating its effectiveness in bridging the gap between simulation and reality.

**Qualitative Results.** We conduct a visual comparison across four representative tasks, as shown in Fig. 4. From left to right, the tasks are *Move Playing-Card Away*, *Ranking Blocks RGB*, *Adjust Bottle*, and *Shake Bottle*. Note that the simulated objects differ from their real-world counterparts in appearance. For instance, the robotic gripper is black in reality rather than gray in the simulator, the dominant color of the playing card's surface pattern is white instead of blue, and the Coca-Cola bottle cap is yellow rather than red. As shown in the figure, the zero-shot model fails to capture the essence of sim-to-real transfer, it creates a superficial "realism" by exaggerating color saturation or smoothing surface textures—likely a side effect of training data dominated by human face images—leading to severe hallucination artifacts. In contrast, our method targets the critical discrepancies between simulation and reality. It faithfully reproduces surface attributes such as color and material (e.g., the reflective plastic of the Fanta bottle) as well as internal dynamics (e.g., visible liquid motion when shaking the Coca-Cola bottle), producing results much closer to real-world observations and thereby validating the effectiveness of our approach for sim-to-real data generation.

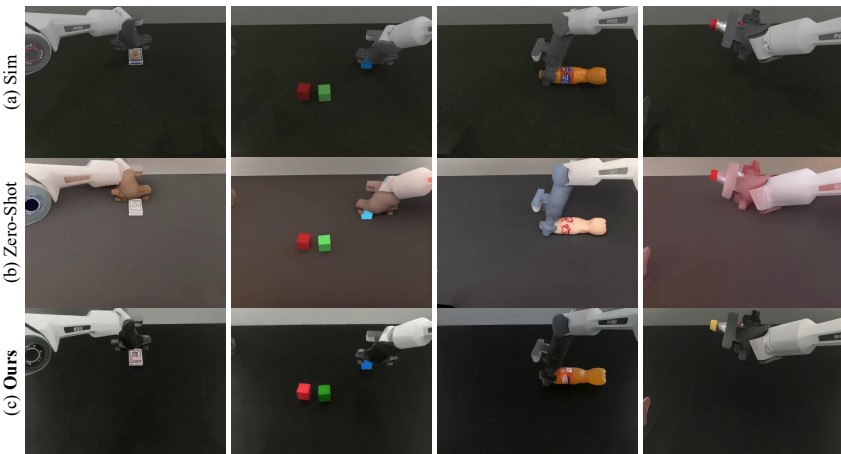

Figure 4: Visual comparison of generated results across four different tasks. Rows correspond to: (a) **Sim**: classical simulation results without neural refinement, (b) **Zero-Shot**: outputs from the untuned base model, and (c) **Ours**: pseudo-realistic videos produced by our neural simulation method.

### 4.2 REAL WORLD EXECUTION WITH COMPOSITIONAL WORLD SIMULATION

**Baselines.** To rigorously quantify the benefit of our proposed compositional world simulation pipeline under an extremely limited real-world demonstration budget, we trained six instances of Diffusion Policy (DP) Chi et al. (2023) according to the following data-mixture regimes: 1) 10 Real: learning solely from 10 real-world demonstrations. 2) 20 Real: doubling the real-world budget to 20 demonstrations to isolate the gain of additional real-world data. 3) 200 Sim Pretrain + 10 Real: pre-training on 200 RoboTwin-simulated demonstrations followed by fine-tuning on the same 10 real-world demonstrations used in Regime 1. 4) 10 Real + 200 Sim: jointly training on the 200 RoboTwin-simulated and 10 real-world demonstrations from scratch. All demonstrations used here are same as Regime 3. 5) 10 Real + 200 Pseudo-Real: jointly training on the 200 pseudo-real demonstrations, which were generated by our compositional world simulation pipeline previously,

Table 2: Real-world evaluation on *Shake Bottle*, *Move Playing-Card Away*, and *Stack Blocks Two*. The compared methods include DP trained under six data-mixture regimes: *10 Real*, *20 Real*, *200 Sim Pretrain + 10 Real*, *10 Real + 200 Sim*, *10 Real + 200 Pseudo Real*, and *200 Pseudo Real*. "OOD" abbreviates out-of-distribution.

| Real World Task | Spatial Distribution | 10 Real | 20 Real | 200 Sim Pretrain + 10 Real |
|---|---|---|---|---|
| *Shake Bottle* | In Domain | 10/30 | 25/30 | 12/30 |
| | OOD | 0/30 | 1/30 | 0/30 |
| *Move Playing-Card Away* | In Domain | 12/30 | 24/30 | 15/30 |
| | OOD | 0/30 | 0/30 | 2/30 |
| *Stack Blocks Two* | In Domain | 5/30 | 13/30 | 8/30 |
| | OOD | 0/30 | 0/30 | 0/30 |

| Real World Task | Spatial Distribution | 10 Real + 200 Sim | 10 Real + 200 Pseudo Real | 200 Pseudo Real (Zero Shot) |
|---|---|---|---|---|
| *Shake Bottle* | In Domain | 8/30 | **28/30** | 10/30 |
| | OOD | 0/30 | **12/30** | 5/30 |
| *Move Playing-Card Away* | In Domain | 10/30 | **29/30** | 12/30 |
| | OOD | 1/30 | **17/30** | 9/30 |
| *Stack Blocks Two* | In Domain | 2/30 | **15/30** | 7/30 |
| | OOD | 0/30 | **6/30** | 3/30 |

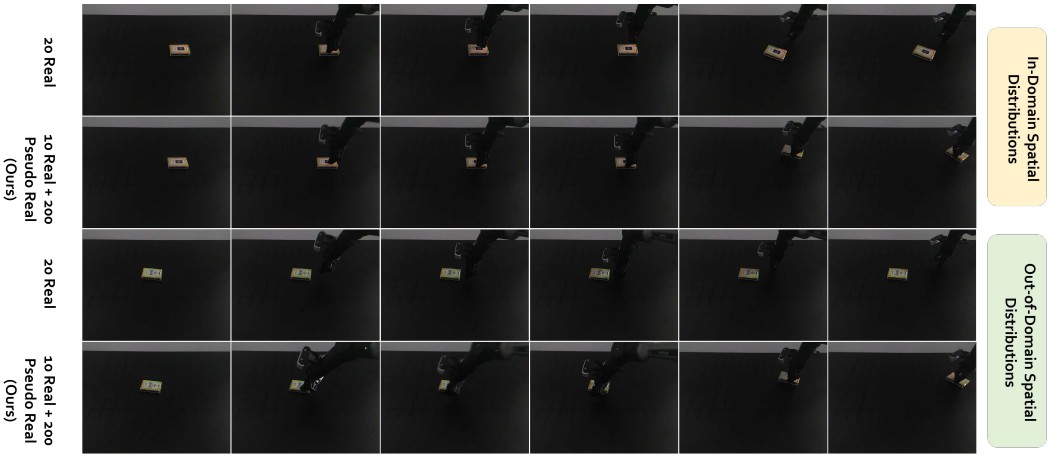

Figure 5: Visualization of DP performance on *Move Playing-Card Away*. Top two rows: objects lie initially within the region predefined in collected real-world demonstrations (in-domain spatial distribution). Bottom two rows: initial positions are outside the region (out-of-domain spatial distribution). Policies shown are trained under *20 Real* and *10 Real + 200 Pseudo Real*, respectively.

and 10 real demonstrations used in Regime 1 from scratch. 6) 200 Pseudo-Real (Zero-Shot): zero-shot training exclusively on the 200 pseudo-real demonstrations used in Regime 5, establishing an upper-bound on the performance achievable by DP without any real-world supervision.

**Quantitative Results.** As shown in Tab. 2, DP performs poorly when only 10 real-world demonstrations are available (cf. 10 Real), and its success rate improves steadily as more real-world data are provided (cf. 20 Real), underscoring the importance of sufficient real-world experience. Simulated data from the traditional simulator (such as RoboTwin) also helps, yet the benefit is capped by the visual–physical gap between real-world and simulated environments (cf. 200 Sim Pretrain + 10 Real and 10 Real + 200 Sim). In contrast, the Pseudo-Real demonstrations generated by our compositional world simulation pipeline narrow this gap and yield a substantial increase in task success rate (cf. 10 Real + 200 Pseudo Real), and even delivering non-trivial performance in the complete absence of real data (200 Pseudo Real).

Table 3: Quantitative evaluation of DP new-object generalization across six data mixtures.

| Real World Task | 10 Real | 20 Real | 200 Sim Pretrain + 10 Real |
|---|---|---|---|
| *Shake Bottle* | 0/30 | 0/30 | 0/30 |
| *Move Playing-Card Away* | 1/30 | 2/30 | 1/30 |

| Real World Task | 10 Real + 200 Sim | 10 Real + 200 Pseudo Real | 200 Pseudo Real (Zero Shot) |
|---|---|---|---|
| *Shake Bottle* | 0/30 | **15/30** | 9/30 |
| *Move Playing-Card Away* | 0/30 | **21/30** | 11/30 |

## 4.3 GENERALIZATION

To further validate the fidelity of our pipeline in reproducing real-world scenarios, we conducted an ablation study on the ability of DP to generalize to new spatial layouts and new objects. All DP evaluated here are identical to those introduced in Sec. 4.2.

**Generalization to Novel Spatial Distributions.** It is necessary to note the initialized position of every object in the collected real-world demonstrations was confined to a predefined limited region (see Sup. C.2.1). At inference, we relocated the same objects to previously unseen regions and recorded the success rates of all DPs. Tab. 2 and Fig. 5 shows that DPs trained solely on real data exhibit almost zero generalization to the new regions; the spatial diversity present in traditional RoboTwin simulations is likewise rendered ineffective by the sim-to-real gap, yielding no measurable improvement. In contrast, the pseudo-real demonstrations generated by our compositional world simulation pipeline consistently lift performance across the relocated configurations, confirming that the synthesized scenes faithfully reproduce the spatial statistics of the real-world environments.

**Generalization to New Objects.** We evaluate shape- and color-level generalization by substituting new objects at inference time. Concretely, in the real-world demonstrations we employ a Fanta bottle and a blue playing card, and at inference time these are replaced by other bottles (i.e. Coca-Cola, Sprite and Nongfu Spring Oriental Leaf Tea) and a red playing card, respectively. As shown in Table 3 and Fig. 6, simulated demonstrations collected from RoboTwin bring no improvement in the generalization to new objects, whereas the Pseudo-Real demonstrations generated by our compositional world simulation pipeline yield a clear boost in success rate. This demonstrates that our method preserves real-world properties and supports transfer to unseen objects.

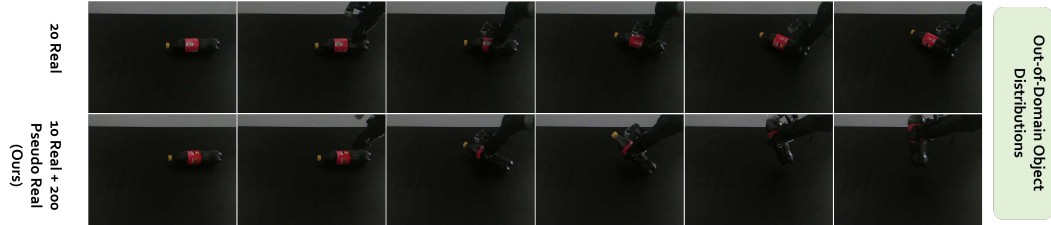

Figure 6: Generalization visualization of DP on *Shake Bottle* under out-of-domain object distributions. Top: policy trained with *20 Real*. Bottom: policy trained with *10 Real + 200 Pseudo Real*.

## 5 CONCLUSION

We presented *Compositional Simulation*, a hybrid framework that integrates classical and neural simulation through a real–sim–real pipeline to generate accurate and consistent action–video pairs. Our approach leverages limited real-world data to create large-scale, diverse training datasets, substantially narrowing the sim2real domain gap. Experiments show that *Compositional Simulation* improves real-world policy success rates and enables stronger generalization across tasks, spaces, and objects. This work offers a scalable path toward robust data generation for embodied intelligence and opens avenues for extending to richer modalities and broader robotic embodiments.

**Limitation and Future Work.** Our experiments focus on tabletop manipulation, though the framework could be extended to more complex embodiments such as mobile manipulation with wheeled robot. And no specialized design was introduced for the neural simulator. Future work may investigate stronger action conditioning for improved action–video consistency and the use of unpaired data to enhance capability and generalization, which would further advance compositional simulation.

ETHICS STATEMENT

The research reported in this paper involves only standard robotic manipulation of everyday objects (such as beverage bottles, playing-cards, blocks) in a laboratory setting. No human or animal subjects, personal data, sensitive information, or hazardous materials were involved. All experiments were conducted in compliance with the safety regulations of the host institution and the relevant technical guidelines for robotic systems.

REPRODUCIBILITY STATEMENT

To facilitate full reproducibility, we provide:

1. Complete source code for data collection, compositional world simulation, model training, and evaluation at github.

2. Detailed hyper-parameters and network architectures in Appendix C.

3. Comprehensive documentation of the real-world platform and evaluation protocol in Appendix D.

4. Video recordings of every real-world trial, together with the corresponding RGB-D sensor streams, which will be made publicly available upon acceptance.

All experiments were conducted on the open-source RoboTwin simulator and our standardized robotic platform; containerized environments and exact dependency versions are released to guarantee bitwise reproducibility.

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

# APPENDIX

## A  USE OF LLMs

This paper was written by the authors without any generative contribution from large language models (LLMs). LLMs were employed solely for language polishing and grammatical refinement; no scientific content, technical claims, or novel interpretations were produced or altered by these tools.

## B  TASK DETAILS

To facilitate assets alignment between the real-world and simulated settings, we select three representative tasks in RoboTwin Chen et al. (2025)—*Shake Bottle*, *Move Playing Card Away*, and *Stack Blocks Two*—to evaluate our compositional world simulation framework. Their respective success criteria are defined as follows.

- *Shake Bottle* involves four beverages—Fanta, Coca-Cola, Sprite, and Nongfu Spring Oriental Leaf Tea. Among these beverages, Fanta is employed to collect real-world demonstrations, while the remaining three serve as new objects for an ablation study on model generalization. The task is deemed successful if the robot grasps the bottle from the desktop, lifts it to a predefined height, and performs a shaking motion.

- *Move Playing-Card Away* employs two types of playing cards that differ in color—blue and red. Following the same protocol as *Shake Bottle*, the blue playing card is used to construct the real-world training dataset, whereas the red playing card serves as an unseen object for evaluating model generalization. The task is considered successful once the robot grasps the designated card and transports it completely away from the central region of the desktop.

- *Stack Blocks Two* utilizes two colored blocks—green and yellow. This task is designed primarily to assess the model's ability to generalize to novel spatial configurations. Success is achieved when the robot first places the green block at the designated position and subsequently stacks the yellow block precisely on top of it.

## C  TRAINING DETAILS

### C.1  NEURAL SIMULATOR TRAINING DETAILS

As mentioned in Sec. 4.1, our Neural Simulator builds upon Stable Diffusion 1.5 Rombach et al. (2022), a state-of-the-art latent text-to-image diffusion model capable of generating high-fidelity visual content from textual prompts. We provide a fixed sim-to-real instruction as its text input, namely: *"Change the image style from the image style of the simulated environment to the image style captured by a DSLR camera."*. Next, we pair the initial simulation data produced by our Classical Simulator with corresponding real-world data to form simulation–real data pairs. The base model is then fine-tuned on these pairs by minimizing the diffusion model's denoising loss. Finally, an online inference strategy FRESCO Yang et al. (2024) is applied to the fine-tuned model to generate the final high-quality pseudo-realistic videos.

All experiments are conducted on one NVIDIA H200 GPU. During fine-tuning, the video data is first converted into image sequences at 10 FPS and organized into a training set, with $1/5$ of the data randomly sampled as a validation set. The model is trained for 30 epochs with a batch size of 8 and a gradient accumulation step of 4, taking approximately 10 hours. We employ the `torch.optim.AdamW` optimizer with a learning rate of $5.0 \times 10^{-5}$ and a linear warm-up ratio of 0.01. For the loss function, the diffusion model's denoising loss is empirically weighted by 1.0, the perceptual loss (measuring feature-level differences between generated and real images) is weighted by 0.2, and the pixel-wise loss (computing the RGB mean squared error between generated and real images) is weighted by 0.1. During inference, we follow the default parameter settings of FRESCO, except that the minimum key-frame sampling interval is set to 3, to ensure smoother video generation.

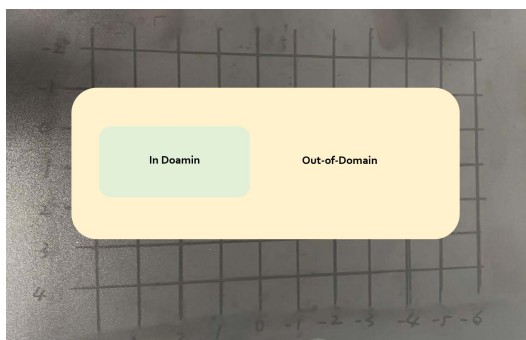 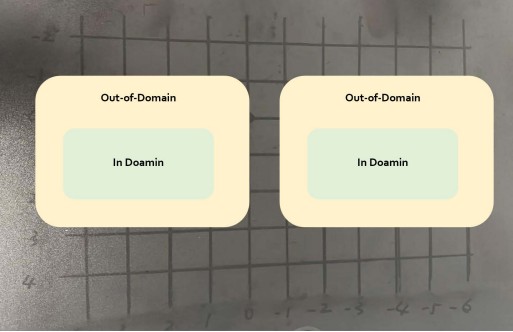

(a) Spatial Distributions of Shake Bottle and Move Playing-Card Away        (b) Spatial Distributions of Stack Blocks Two

Figure 7: Definition of in-domain and out-of-domain spatial distributions in different tasks. Both terms refer exclusively to the initial position of objects before being manipulated. Positions are labeled in-domain if and only if they appear in the collected real-world demonstrations; all others are deemed out-of-domain.

## C.2 DP Training Details

### C.2.1 Demonstrations

**Real-World Demonstrations** were meticulously collected via human teleoperation using a pair of PiPER Teach Pendants (see Sup. D.1). For each task, we recorded only 20 trajectories, all confined to the in-domain spatial and object distribution. Concretely, demonstrations for *Shake Bottle* and *Move Playing-Card Away* were acquired exclusively with the Fanta bottle and the blue playing card starting within the in-domain region illustrated as Fig. 7(a), respectively. And for *Stack Blocks Two*, the green and yellow blocks were always placed in the left and right in-domain zones at the beginning of demonstration collection (Fig. 7(b)). Finally, we randomly selected 10 out of these 20 demonstrations to construct the data-mixture regime *10 Real*, and used all 20 to construct the regime *20 Real*.

**Simulated Demonstrations** were generated in the traditional simulator RoboTwin, and we collected 200 trajectories for each task. In contrast to the real-world ones, these simulated demonstrations deliberately incorporated out-of-domain spatial arrangements and objects. Specifically, for *Shake Bottle* we used not only the Fanta bottle but also Coca-Cola, Sprite and Nongfu Spring Oriental Leaf Tea, while for *Move Playing-Card Away* we included the red playing card in addition to the blue one. Besides, all objects might be placed in out-of-domain regions when data collection started. All 200 simulated demonstrations were employed to construct the data-mixture regimes *10 Real + 200 Sim* and *200 Sim Pre-train + 10 Real*.

**Pseudo-Real Demonstrations** were produced by our compositional world-simulation framework under the same out-of-domain spatial and object settings employed for Simulated Demonstrations. The full set of 200 pseudo-real trajectories was used to establish the data-mixture regimes *10 Real + 200 Pseudo-Real* and *200 Pseudo-Real*.

### C.2.2 Training Settings

We use Diffusion Policy (DP) Chi et al. (2023), a generative method based on imitation learning. We employ a CNN-based Diffusion Policy as the backbone of our visuomotor model. The prediction horizon is set to 8, with 3 observation steps and 6 action steps. For data loading, we use a batch size of 256. The optimizer is `torch.optim.AdamW` with a learning rate of $1.0 \times 10^{-4}$, betas in $[0.95, 0.999]$, and $\epsilon = 1.0 \times 10^{-8}$. A learning-rate warmup is applied for the first 500 steps, followed by 300 training epochs for all benchmark tasks.

Each policy is trained independently on a single NVIDIA H200 GPU for 300 epochs. As a reference, using a dataset of roughly 200 demonstration episodes (average length $\approx 300$), training a single policy for 300 epochs takes about 20 hours.

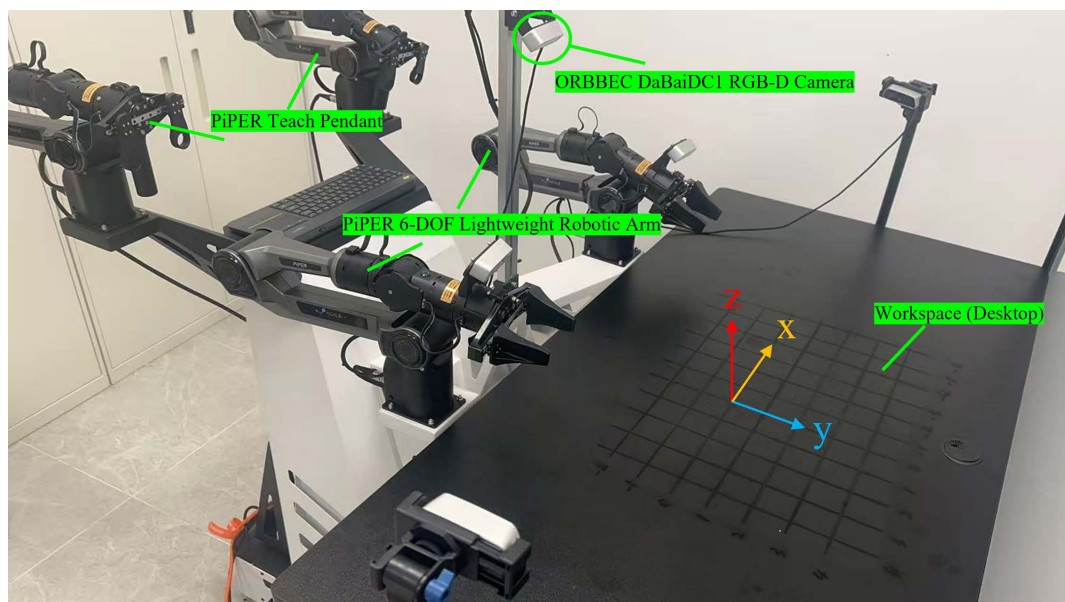

Figure 8: Real-world evaluation platform.

# D EVALUATION DETAILS

## D.1 PLATFORM

As depicted in Fig. 8, the real-world evaluation were performed with two ORBBEC PiPER 6-DOF Lightweight Robotic Arms, each equipped with a two-finger gripper (maximum opening 70 mm, gripping force 40 N). A fixed, top-down ORBBEC DaBaiDC1 RGB-D camera provided a global RGB view of the workspace, defined as the central area of a black tabletop. In addition, a pair of PiPER Teach Pendants enabled teleoperation of the arms, allowing efficient collection of real-world demonstrations. All hardware units were connected to a workstation housing an NVIDIA GeForce RTX 4090 GPU, which stored the captured observations, performed model inference, issued control commands, and drove the robotic arms in real time.

## D.2 EVALUATION SETTINGS

For each ablation dimension—in-domain spatial/object configurations, out-of-domain spatial layouts, and out-of-domain objects—we independently constructed a fixed set of 30 diverse real-world trials. Every policy trained under a different data-mixture regime was evaluated on the corresponding 30-trial split, guaranteeing that all comparisons within a distribution are performed on an identical test bed. Task-success criteria are provided in Sup. B.

# E SIM2REAL NEURAL SIMULATION DETAILS

As stated in Sec. 3.2, we enforce strict alignment between the real-world and simulated environments—encompassing background and object appearance, camera intrinsics/extrinsics, and object positions—to enable effective Sim2Real neural simulation. To prevent any learning-induced errors from propagating into the subsequent training of the neural simulator, we adopt a purely rule-based alignment pipeline rather than a data-driven one. Concretely, we first parameterize the relevant attributes of the real-world scene and then transfer the estimated parameters to configure the simulated environment accordingly.

### E.1 BACKGROUND AND OBJECT ALIGNMENT

**Background Alignment** mainly parameterizes both the visual appearance of the desktop and the laboratory walls. Using the fixed RGB-D camera described in Sec. D.1, we first capture images of the table surface and the wall regions. A digital color-picker is then applied to the acquired images to extract representative RGB values.

**Regular-Object Alignment** covers primitives such as blocks, spheres, and cylinders whose geometry can be described by a small set of metric dimensions. For these instances, we first measure their principal axes (length, width, height, diameter, etc.) with calipers. Their Appearance are parameterized by acquiring an orthographic RGB patch of the object's most representative face and extracting the median albedo via a color-picker tool—no additional texture map is required, yielding a compact, error-tolerant representation.

**Special-Object Alignment.** Owing to RoboTwin's one-to-one digital twins of real-world assets—including Finda, Fanta, Coca-Cola, Sprite, Nongfu Spring Oriental Leaf Tea bottles and the playing cards—we can directly pair every physical item with its pre-modeled, dimension- and texture-matched counterpart. This eliminates the need for on-the-fly scanning or manual modeling: each real-world bottle or card is simply mapped to its pre-registered URDF/FBX model, guaranteeing sub-millimetre geometric agreement and pixel-level texture consistency between reality and simulation.

### E.2 CAMERA CALIBRATION AND ALIGNMENT

Camera parameterization focuses on retrieving its intrinsic and extrinsic. The intrinsics can be known directly from its technical documentation; hence, the following section details only the extrinsic-calibration pipeline employed in our setup.

To ensure consistency with RoboTwin, we establish a real-world coordinate system as illustrated in Fig. 8. Within this coordinate system, we first place a calibration checkerboard and obtain the 3D coordinates of its corner points. Subsequently, we capture images using the mounted camera and extract the 2D pixel coordinates of those checkerboard corners via corner detection. The code implementing this procedure is listed below.

```python
import cv2
import numpy as np

# --------------------------
PATTERN_SIZE = (7, 4)          # Number of checkerboard corners along rows and columns
IMG_PATH     = 'chess.jpg'     # Path of the captured image
# --------------------------

img  = cv2.imread(IMG_PATH)
if img is None:
    raise FileNotFoundError(IMG_PATH)
gray = cv2.cvtColor(img, cv2.COLOR_BGR2GRAY)

# 1. Checkerboard Detection
ret, corners = cv2.findChessboardCorners(
    gray, PATTERN_SIZE,
    cv2.CALIB_CB_ADAPTIVE_THRESH + cv2.CALIB_CB_FAST_CHECK + cv2.CALIB_CB_NORMALIZE_IMAGE)
if not ret:
    raise RuntimeError('Checkerboard detection failed!')

corners = cv2.cornerSubPix(
    gray, corners, (11, 11), (-1, -1),
    criteria=(cv2.TERM_CRITERIA_EPS + cv2.TERM_CRITERIA_MAX_ITER, 30, 0.001))
pts2d = corners.reshape(-1, 2).tolist()

# 2. Mouse Callback
def on_mouse(event, x, y, flags, param):
    global pts2d, img_show
    if event == cv2.EVENT_LBUTTONDOWN:
        idx = min(range(len(pts2d)),
                    key=lambda i: (pts2d[i][0] - x) ** 2 + (pts2d[i][1] - y) ** 2)
        if (pts2d[idx][0] - x) ** 2 + (pts2d[idx][1] - y) ** 2 < 400:
            del pts2d[idx]
            redraw()

def redraw():
    global img_show
```

```
38        img_show = img.copy()
39        for idx, (u, v) in enumerate(pts2d):
40            u, v = int(u), int(v)
41            cv2.circle(img_show, (u, v), 5, (0, 0, 255), -1)
42            cv2.putText(img_show, str(idx), (u + 10, v - 10),
43                        cv2.FONT_HERSHEY_SIMPLEX, 0.5, (0, 0, 255), 1, cv2.LINE_AA)
44        cv2.imshow('interactive', img_show)
45
46 # 3. Main Loop
47 cv2.namedWindow('interactive', cv2.WINDOW_NORMAL)
48 cv2.setMouseCallback('interactive', on_mouse)
49 redraw()
50
51 print('Usage:')
52 print('  Left-click on a corner -> toggle delete/undelete')
53 print('  Press q -> save current pts2d.txt and exit')
54 print('  Press r -> restore all originally detected corners')
55 print('  Close the window (x) to quit without saving')
56
57 while True:
58     if cv2.getWindowProperty('interactive', cv2.WND_PROP_VISIBLE) < 1:
59         break
60
61     key = cv2.waitKey(30) & 0xFF  # 30 ms timeout to prevent freezing
62     if key == ord('q'):
63         np.savetxt('pts2d.txt', np.array(pts2d), fmt='%.6f')
64         print('Saved pts2d.txt with', len(pts2d), 'points.')
65         break
66     elif key == ord('r'):
67         pts2d = corners.reshape(-1, 2).tolist()
68         redraw()
69
70 cv2.destroyAllWindows()
```

With the obtained 3D-to-2D correspondences, the camera's extrinsic parameters—position and orientation—can be recovered by solving a Perspective-n-Point (PnP) problem. The implementation is given below.

```
1  import numpy as np
2  import cv2
3
4  # 1. Given Known Intrinsics
5  K = np.array([
6      [488.8112487792969, 0.0, 317.05938720703125],
7      [0.0, 488.8112487792969, 217.4825439453125],
8      [0.0, 0.0, 1.0]
9  ], dtype=np.float64)
10
11 dist = np.zeros((4, 1))  # Distortion Coefficients; set to zero if no distortion.
12
13 # 2. Input
14 pts3d = np.loadtxt('pts3d.txt')   # N×3, World Coordinates
15 pts2d = np.loadtxt('pts2d.txt')   # N×2, Pixel Coordinates
16
17 assert pts3d.shape[0] == pts2d.shape[0], 'Mismatch in point count!'
18
19 # 3. Estimating Extrinsic
20 ok, rvec, tvec = cv2.solvePnP(
21     pts3d.astype(np.float64),
22     pts2d.astype(np.float64),
23     K, dist, flags=cv2.SOLVEPNP_ITERATIVE
24 )
25 if not ok:
26     raise RuntimeError('Solve PnP failed! Please verify that the point correspondences are
27                         correct.')
28 # Rotation Vector -> Rotation Matrix
29 R, _ = cv2.Rodrigues(rvec)
30
31 R_w2c = R
32 t_w2c = tvec.ravel()
33 R_c2w = R_w2c.T
34 t_c2w = -R_c2w @ t_w2c   # 3×1
35
36 position = t_c2w.tolist()
37
38 forward = R_c2w[:, 2]
39 left    = -R_c2w[:, 0]
40 up      = -R_c2w[:, 1]
```

```
41
42 print("static_camera_list:")
43 print("  - name: head_camera")
44 print("    type: D435")
45 print(f"    position: [{position[0]:.3f}, {position[1]:.3f}, {position[2]:.3f}]")
46 print(f"    forward:  [{forward[0]:.3f}, {forward[1]:.3f}, {forward[2]:.3f}]")
47 print(f"    left:     [{left[0]:.3f}, {left[1]:.3f}, {left[2]:.3f}]")
```

### E.3 OBJECT POSITION ALIGNMENT

We emphasize that the real-world frame depicted in Fig. 8 coincides exactly with the world frame employed in RoboTwin. Under this frame, the desktop surface is tessellated into a uniform 5 cm × 5 cm lattice. Object placement is thereby reduced to aligning the object's center of mass with a lattice node; orientation is selected from a prescribed, rule-based catalogue—namely, axis-aligned poses or rotations of 30°, 45°, and 60° about the x- or y-axis. While this discrete parameterization is admittedly naive, it routinely delivers positional errors below one centimetre and angular errors below one degree. A data-driven, continuous 6-DoF alignment module will be investigated in future work to supersede this manual gridding scheme.

## F MORE RESULT VISUALIZATION

### F.1 VISUALIZATION OF GENERALIZATION ON NEW OBJECTS

To highlight the enhanced generalization of DP enabled by our compositional world simulation pipeline, we provide trajectory visualizations of the *Move Playing-Card Away* task in Fig. 6, with additional examples in Fig. 9.

### F.2 VISUALIZATION OF REAL2SIM ALIGNMENT

As detailed in Sup. E, we performed exhaustive Real2Sim alignment. Here we illustrate the final alignment quality for the tasks *Move Playing-Card Away*, *Ranking Blocks RGB*, *Stack Blocks Three* and *Stack Blocks Two* in Fig. 10 and Fig. 11.

### F.3 VISUALIZATION OF SIM2REAL NEURAL SIMULATION

To dynamically demonstrate the effectiveness of our approach in sim-to-real transfer, we further present a visual comparison between the pseudo-realistic videos generated by our Neural Simulator and the initial simulation videos, as shown in Fig. 12. In addition to the tasks included in the main text—*Adjust Bottle*, *Moving PlayingCard Away*, and *Ranking Blocks RGB*—we also consider the *Stack Blocks Three* task. The results indicate that our method consistently maintains strong temporal coherence and perceptual realism throughout the video sequences.

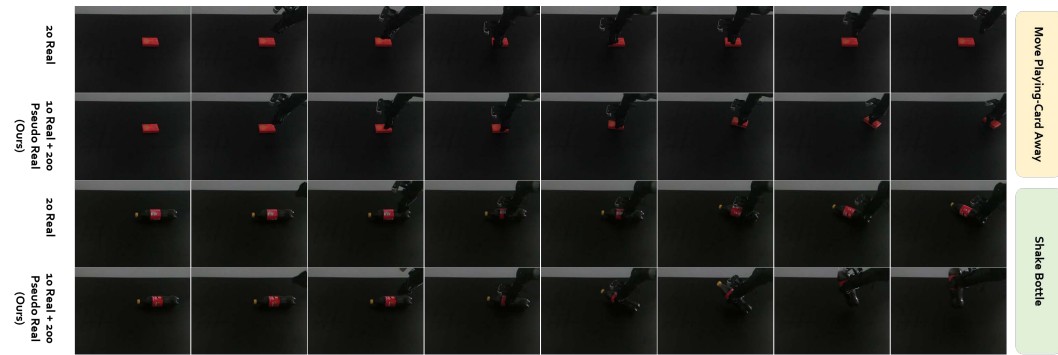

Figure 9: Generalization visualization of DP on new objects. The top two rows are corresponds to *Move Playing-Card Away*, and the bottom two rows are corresponds to *Shake Bottle*. respectively.

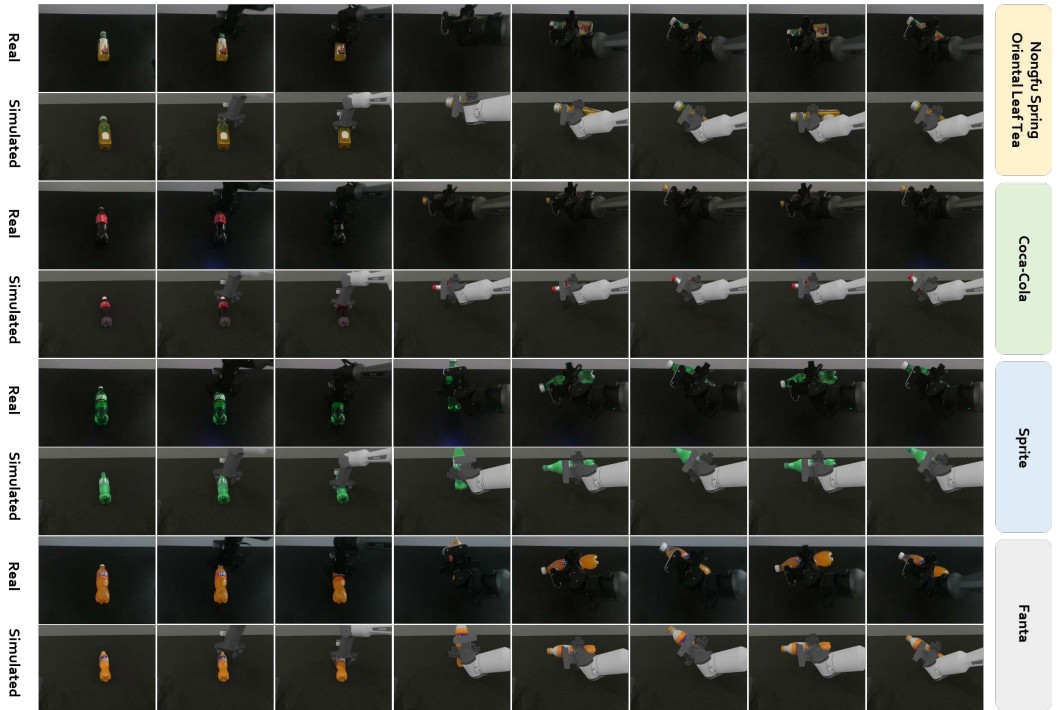

Figure 10: Real2Sim alignment on *Move Playing-Card Away*. From top to bottom: Nongfu Spring Oriental Leaf Tea, Coca-Cola, Sprite, and Fanta.

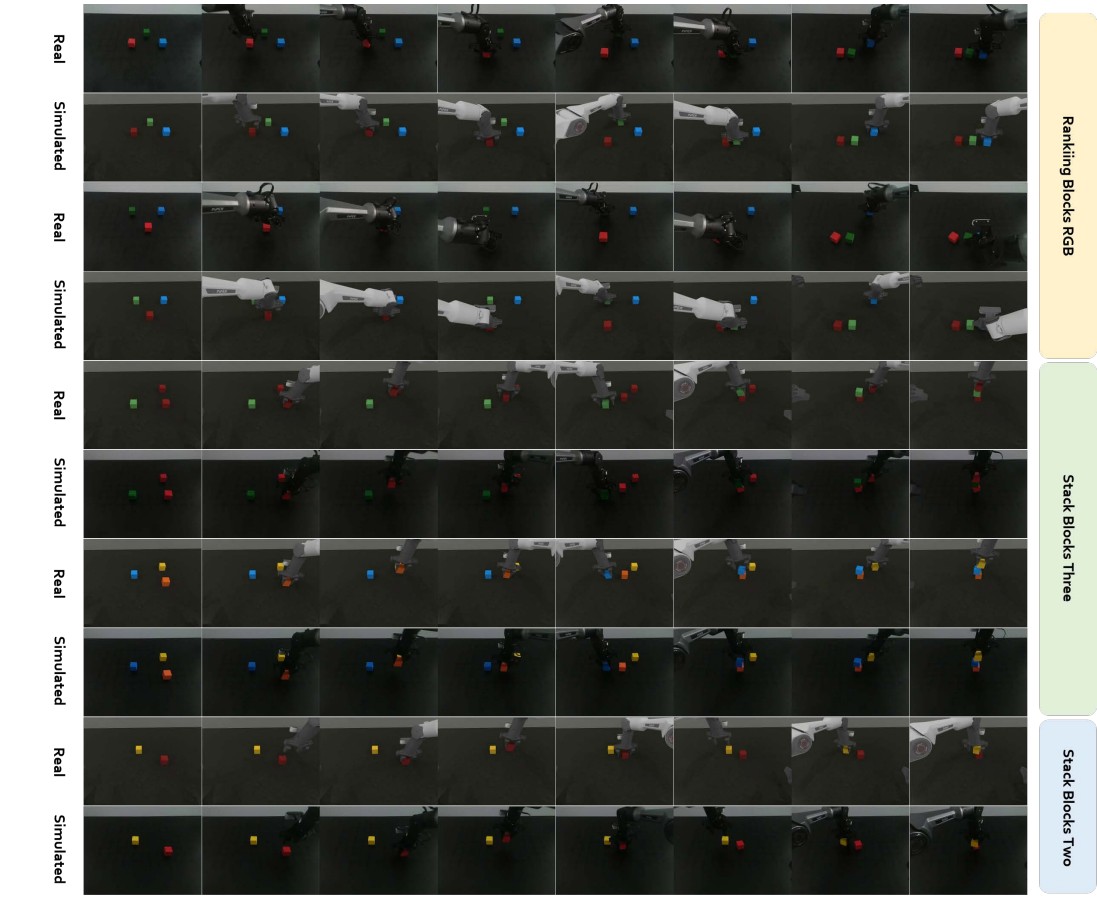

Figure 11: Real2Sim alignment on additional tasks. From top to bottom: *Ranking Blocks RGB*, *Stack Blocks Three* and *Stack Blocks Two*.

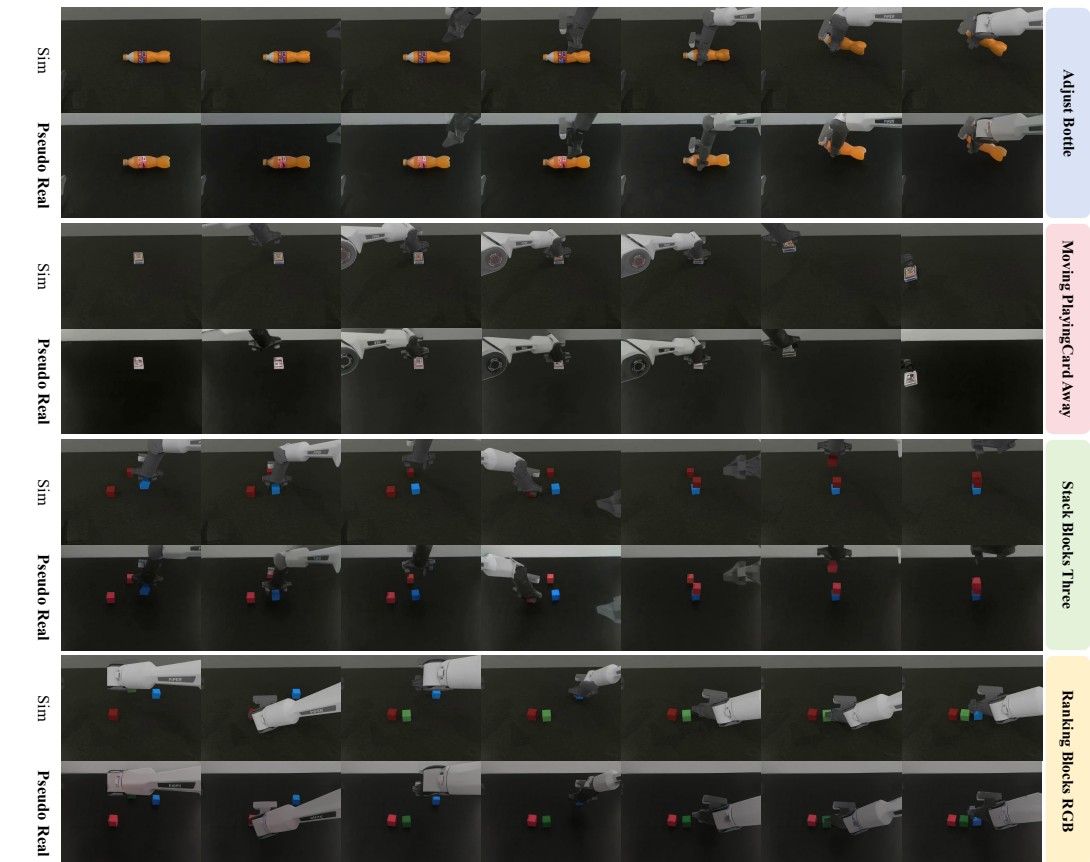

Figure 12: Sim2Real visualization on various tasks. From top to bottom: *Adjust Bottle*, *Moving PlayingCard Away*, *Stack Blocks Three* and *Ranking Blocks RGB*.

