# OpenReview forum: "Building Scalable Real-World Robot Data Generation via Compositional Simulation"
_ICLR.cc/2026/Conference — ICLR 2026 Conference Withdrawn Submission_

### Official Review · Reviewer_U711 · 2025-10-27

**Soundness:** 1
**Presentation:** 2
**Contribution:** 1
**Rating:** 2
**Confidence:** 5

**Summary:**

This submission introduces ComSim, a hybrid framework that integrates classical and neural simulation through a real–sim–real pipeline to generate robot data. Experiments show that the proposed method improves real-robot success rates, and demonstrate generalization capability. However, several concerns exist.

**Strengths:**

- The submission is well-written and easy to follow.

**Weaknesses:**

- The overall contribution of this submission is weak. While authors claim a new real-sim-real pipeline, there exists many real-sim-real works that achieve better performance, such as Chen et al., 2024; Torne et al., 2024; Dai et al., 2024. Using diffusion models for augmentation is also not new. People have already explored this for robotics, such as Yu et al., 2023. Furthermore, the conclusion drawn from this submission---real-robot performance can benefit from synthetic data---is also not new, as researchers have derived from, e.g., Maddukuri et al., 2025.

- The claim about poor sim-real joint training performance (L47) is factually wrong. In fact, researchers have shown that sim-real joint training can significantly help real-robot performance (Maddukuri et al., 2025).

- While authors claim the proposed method addresses issues such as scaling, appearance, physics, and action consistency (Fig. 1), there is no experimental results that support each of this claim.

- For robots to work in complex real-world environments, only visual observations are insufficient. The proposed method is incapable of handling other important modalities, such as proprioception, force, etc.

- While authors claim the proposed method can generate diverse data. Upon a closer look, all data is merely on a tabletop setting with single or only a few objects. The object diversity is also extremely limited.

- The experimented robotic tasks are too simple and trivial. It's merely a tabletop setting with stationary robot arms. Objects are few and quite limited in categories.

- If the proposed method replays the same robot trajectories from the real-world data, given the same amount of seeding real-world data, it's unlikely the trained policies can exhibit better spatial generalization.

- Comparisons to other robot data synthesis methods are necessary, such as MimicGen (Mandlekar et al., 2023), DemoGen (Xue et al., 2025), etc.

## References
- Chen et al., URDFormer: A Pipeline for Constructing Articulated Simulation Environments from Real-World Images, RSS 2024.
- Torne et al., Reconciling Reality through Simulation: A Real-to-Sim-to-Real Approach for Robust Manipulation, RSS 2024.
- Dai et al., Automated Creation of Digital Cousins for Robust Policy Learning, CoRL 2024.
- Yu et al., Scaling Robot Learning with Semantically Imagined Experience, arXiv 2023.
- Maddukuri et al., Sim-and-Real Co-Training: A Simple Recipe for Vision-Based Robotic Manipulation, RSS 2025.
- Mandlekar et al., Mimicgen: A data generation system for scalable robot learning using human demonstrations, CoRL 2023.
- Xue et al., DemoGen: Synthetic Demonstration Generation for Data-Efficient Visuomotor Policy Learning, RSS 2025.

**Questions:**

See above.

---

### Official Review · Reviewer_7ZeK · 2025-10-31

**Soundness:** 2
**Presentation:** 2
**Contribution:** 2
**Rating:** 4
**Confidence:** 3

**Summary:**

This paper proposes learning a neural renderer that maps simulator video–action pairs into realistic videos, thereby augmenting robot manipulation data and reducing the sim-to-real gap. To achieve this, the authors build a simulation environment that closely mirrors the real-world setup, enabling the construction of sim-video / real-video / action triplets. Experimental results demonstrate that the proposed method effectively transforms unseen simulated videos into realistic ones, leading to improved policy performance.

**Strengths:**

1. The idea of leveraging generative AI techniques to mitigate the sim-to-real gap is novel and insightful. It provides a fresh perspective on how modern generative models can be applied to robotics and data augmentation.

2. The paper conducts extensive experiments, offering strong empirical validation for this approach and demonstrating its effectiveness across various scenarios.

**Weaknesses:**

1. The proposed method relies on paired sim-to-real data, which limits scalability. In many real-world applications, it may not be feasible to build a simulator that perfectly matches each real system. The authors should consider exploring unsupervised or unpaired approaches to enable broader generalization and application.

2. The experimental section lacks comprehensive baselines. In particular, since the method assumes the availability of a simulator, it would be beneficial to compare with other simulation-based data augmentation techniques, such as MimicGen, to convincingly establish performance advantages.

**Questions:**

1. Can this method generalize more broadly — for example, would it still work effectively in scenarios where no simulator is available?

2. Could the authors provide stronger baselines to verify the effectiveness of this method compared to other native sim-to-real or data-augmentation approaches?

---

### Official Review · Reviewer_AVmY · 2025-10-31

**Soundness:** 2
**Presentation:** 2
**Contribution:** 3
**Rating:** 2
**Confidence:** 4

**Summary:**

This paper proposes Compositional Simulation, a real–sim–real data generation pipeline that combines classical simulation with a fine-tuned diffusion-based video-to-video model to generate pseudo-real robot demonstrations. A small set of real trajectories are replicated in simulation to train a sim-to-real neural renderer, which is then used to convert large amounts of simulated rollouts into realistic data. Experiments on tabletop tasks show improved real-world performance and stronger spatial and object generalization under low-data regimes.

**Strengths:**

- **Clear motivation and useful problem scope:** Addresses the practical challenge of scaling robot training with minimal real-world data.
- **Simple yet effective hybrid approach:** Real–sim–real pipeline is intuitive and avoids hallucination issues common in video-generation action simulators.
- **Empirical gains:** Demonstrates improvement in both in-domain and OOD settings for real robot manipulation tasks.
- **Data efficiency:** Achieves strong results using only 10 real demos + pseudo-real rollouts, showing value for low-resource robotics.
- **Reproducibility:** Provides training details, simulation setup, and evaluation protocols, facilitating replication.

**Weaknesses:**

1. **Citation formatting inconsistencies** — Please adopt proper LaTeX citation conventions (e.g., `\citep{}`) for clarity and consistency with ICLR standards.
2. **Unaddressed dynamics gap** — The method primarily tackles appearance transfer; however, real-world dynamics mismatch remains unstudied, limiting applicability to more complex manipulation domains.
3. **Over-simplified visual settings** — Experiments rely on black backgrounds and isolated tabletop objects, which reduces visual difficulty; simpler sim-to-real techniques or classical augmentation pipelines may already perform well under such conditions.
4. **Limited task complexity and diversity** — OOD generalization is demonstrated only on basic tabletop tasks. Evaluating more realistic, cluttered, or multi-stage manipulation scenarios would substantially strengthen the evidence.
5. **Lack of real robot execution videos** — Real-world rollout videos are essential to verify qualitative behavior, demonstrate consistent control performance, and support robustness claims.

**Questions:**

1. **Why not directly edit policy inputs?**
   How does compositional simulation compare to applying image editing or video style-transfer directly on policy observation streams, potentially leveraging large human video corpora? Given your test-time environment setup, one might expect that direct image editing approaches could yield similar benefits — a clarification on this comparison would be valuable.

2. **Include real-image references in qualitative results**
   For clarity and to better assess target realism, please include paired real images alongside the pseudo-real outputs in qualitative figures.

3. **Clarify Figure 2 embedding dimensions**
   Why is the projected latent feature dimension larger than the final reconstructed video representation?

---

### Official Review · Reviewer_Hqoy · 2025-11-01

**Soundness:** 3
**Presentation:** 3
**Contribution:** 3
**Rating:** 4
**Confidence:** 4

**Summary:**

The authors present a method to leverage simulation data directly for learning real world tasks. The approach is primarily based on closing the visual gap between simulation and real world environments. They use a classical simulator (with classical rendering), and train a video model to translate image sequences from simulation to their counterparts in the real world. To train the video model, they carefully collect aligned data in simulation and real world. Experiments show that co-training with a small amount of real world data and simulation data translated to look like real world data, yields superior performance to using real-only, or raw simulation data.

**Strengths:**

- The paper is well-written, easy to follow, and the motivation is clear.
- Experiments in the real world are extensive, including both in domain and out of domain analyses.

**Weaknesses:**

An overall limitation of this work is in its scalability:
- First, building simulations for arbitrary real-world tasks is challenging, due to issues with modeling replicas of complex scenes, and deformable manipulation.
- Another limitation is the requirement of collecting aligned data between simulation and the real world, which can be very time-consuming and error-prone.

Finally, this work focuses primarily on the visual gap, rather than other potential gaps between simulation and the real world, such as physics and the content modeling gap. The overall impact of this approach may be limited, compared to full neural world models that explicitly model all aspects of simulation (physics and visuals) by learning from real-world data.

**Questions:**

- Are there ways to train the video model without aligning simulation and real-world trajectories?
- How can this approach be extended to support existing large-scale simulation datasets, to close the visual gap for them and make them useful for learning real-world tasks?
- It would be interesting to see a comparison to methods that train on raw real-world and simulation trajectories, trained with heavy domain randomization.
- Would the benefits of co-training with "translated" simulation data be present if training on larger real-world datasets, eg. with 50 real-world demonstrations?

---

### Note · Authors · 2026-01-26

I have read and agree with the venue's withdrawal policy on behalf of myself and my co-authors.

---

### Meta-Review · Area_Chair_BGRV · 2026-01-06

**Summary:**

The reviewers raised substantial concerns about this submission's novelty, experimental scope, and technical approach. Reviewer U711, who expressed the highest confidence, identified fundamental issues with the contribution, noting that real-sim-real pipelines and diffusion-based augmentation for robotics have been extensively explored in recent work. The reviewer also contested a core factual claim in the paper regarding sim-real joint training performance. Reviewers AVmY and 7ZeK both rated the paper as reject, citing limited task complexity, over-simplified experimental settings with black backgrounds and tabletop objects, and missing comparisons to relevant baselines like MimicGen and other simulation-based augmentation methods. The requirement for paired sim-to-real data was identified as a significant scalability limitation. Reviewer Hqoy, while more positive about the writing quality and real-world experiments, acknowledged that the approach's focus on visual gaps rather than physics and dynamics gaps limits its overall impact compared to full neural world models.

**Reviewer Concerns:**

There was no rebuttal provided by the authors.

**Reviewer Scores:**

Given the absence of a rebuttal and discussion, the reviewers would likely have maintained their original assessments.

---

### Decision · Program_Chairs · 2026-01-26

Reject